# Prevalence and Correlates of Prescription Drug Misuse in a Nationwide Population Survey in Taiwan

**DOI:** 10.3390/ijerph182412961

**Published:** 2021-12-08

**Authors:** Shu-Wei Liu, Chia-Yi Wu, Ming-Been Lee, Ming-Chi Huang, Chia-Ta Chan, Chun-Ying Chen

**Affiliations:** 1Department of Psychiatry, Shin Kong Wu Ho-Su Memorial Hospital, Taipei 111, Taiwan; hitdre@gmail.com (S.-W.L.); mbl66784@gmail.com (M.-B.L.); paneth@mail2000.com.tw (C.-T.C.); 2School of Nursing, College of Medicine, National Taiwan University, Taipei 10617, Taiwan; 3Department of Nursing, National Taiwan University Hospital, Taipei 100, Taiwan; 4National Taiwan Suicide Prevention Center, Taipei 100, Taiwan; dianachen410523@yahoo.com.tw; 5Department of Psychiatry, College of Medicine, National Taiwan University, Taipei 10617, Taiwan; 6Department of Addiction Sciences, Taipei City Psychiatric Center, Taipei City Hospital, Taipei 10341, Taiwan; mingchyihuang@gmail.com

**Keywords:** prescription drug misuse, self-rated health, self-efficacy, psychopathology, prevalence, psychosocial correlates, population-based survey

## Abstract

Background: Prescription drug misuse (PDM) is a critical mental health issue relating to psychiatric morbidity. This study investigated the prevalence of PDM and its associated psychopathology and psychosocial factors in the general population in Taiwan. Methods: The survey randomly selected a representative sample >15 year-olds using the stratified proportional randomization method. The measurements included demographic variables, previous experience with PDM, self-rated physical and mental health, health self-efficacy, risk factors for suicidality, and psychological distress. Results: The weighted one-year prevalence of PDM was 8.5% (*n* = 180) among 2126 participants. Those with psychological distress and lifetime suicide ideation (23.3%) or suicide attempts (5.0%) were significantly associated with PDM. PDM was also prevalent among those with poorer self-rated health and lower self-efficacy. Insomnia (OR = 1.52), depression (OR = 1.77), and low self-efficacy (OR = 2.29) had higher odds of PDM after adjustment in the logistic regression model. Conclusions: Individuals who misused prescription drugs had a higher prevalence of psychological distress and suicidality and lower levels of self-rated health. Prescription drug misuse problems should be screened for early prevention when prescribing medications for people with insomnia, depression, or lower perceived health beliefs or conditions.

## 1. Introduction

Prescription drug misuse (PDM) is defined as using a prescription drug in a manner or a dose other than that prescribed, taking someone else’s prescription, or taking medication for feeling euphoric [1]. The most common types of PDM include opioids as analgesics, central nervous system depressants as sedatives, and central nervous system stimulants [2]. PDM is a critical problem that results in many severe medical consequences [3], increased emergency room visits, overdose deaths related to prescription drugs [4,5,6,7,8], and admissions for addiction treatment [1]. For instance, overdose deaths involving prescription opioids were five times higher in 2016 than in 1999 [9]; furthermore, Benzodiazepine-related overdose deaths increased by more than 400% from 1996 to 2013 [10], and emergency department visits for Benzodiazepines increased by over 300% from 2004 to 2011 [6].

PDM is associated with the misuse of other substances and psychiatric comorbidities. Compared with the general population, people with substance use disorders (particularly opioid use disorder) have much higher rates of Benzodiazepine misuse [9]. Benzodiazepine misuse is also common among people who use drugs (e.g., cocaine and amphetamines) and/or misuse other types of prescription drugs [9]. People who misuse prescription drugs have high rates of anxiety, mood, and personality disorders [10]. A lifetime history of Benzodiazepine misuse is associated with an increased risk of psychiatric disorders [11]; combined use of prescription drugs such as Opioid analgesics and Benzodiazepines may increase the possibility of overdose death [6] and higher severity of suicide ideation [2,12]. Furthermore, a longitudinal and nationally representative survey funded by the US National Institute on Alcohol Abuse and Alcoholism indicated that the nonmedical use of prescription drugs is a risk factor for the onset of psychiatric disorders (e.g., depressive disorders, bipolar disorders, and anxiety disorders), and that such non-medical use increases the risk of recurrence of alcohol use disorder and substance use disorder [11]. Examining the association between PDM and suicide ideation may inform preventive strategies to reduce suicide risks.

Studies investigating substance use disorders have emphasized that self-efficacy is an essential predictor of substance use. Previous treatment programs for substance use disorder aimed to promote self-efficacy [13]. Self-efficacy was also identified as a critical predictor of how an individual develops coping behaviors, performance, and perseverance in the face of difficulties [14]. Health-related self-efficacy was defined as an adaptive cognitive factor of perceived health control [15]. It was included in this study because the variable plays a key role in predicting treatment outcomes of and longer abstinence from substance use disorders.

Further, self-efficacy on the belief that one’s own efforts can achieve desired results (e.g., health promotion or psychosocial function) is necessary to sustain idealized coping behaviors and quality of life [15]. The concept implies that higher self-efficacy on personal effort promotes prevention from relapse of poor coping strategies, such as binge-drinking, bulimia, or prescription medication misuse. Therefore, we conjectured that lower self-efficacy results in poorer quality of life among people with PDM, which then influences the individual’s drug use pattern. Previous research has demonstrated that Benzodiazepine misuse and dependence are associated with poor self-reported quality of life, high pain severity, sleep dysfunction, and repeated emergency department visits in various populations [9]. A national survey of substance use in Taiwan in 2014 revealed that people who previously used nonprescription sedatives, hypnotics, or analgesics exhibited more severe depressive moods [16]. Thus, we hypothesized that self-efficacy is associated with both drug use patterns and psychological health.

Although people who misuse prescription drugs often experience severe consequences, they rarely seek professional medical help. In the abovementioned 2014 substance use survey, only 4.4% of those who reported PDM sought medical help [17]. Despite this, policymakers and the scientific community continue to overlook Benzodiazepine misuse [17]. This may be attributable to the low perceived risk associated with Benzodiazepines [17,18] or barriers that prevent people with drug misuse from seeking medical help, such as stigma [19,20]. In order to investigate treatment, this study investigated the prevalence of PDM in a representative sample in Taiwan and examined the psychological factors associated with PDM.

## 2. Materials and Methods

### 2.1. Study Design, Participants, and Procedure

The study participants were recruited through a telephone survey on population mental health conducted by the National Taiwan Suicide Prevention Center (NTSPC). Ethical approval was acquired from the general hospital to which the corresponding author was affiliated with (reference number 201204034RIC). The survey recruited a representative sample of the general population aged 15 years and over in Taiwan. A computer-assisted telephone interview (CATI) system was applied to identify potential respondents using telephone numbers selected using a stratified proportional randomization method according to the distribution of gender, age, and population size in 22 different geographic areas of Taiwan. The target sample size was a priori set as 1600. The number of participants was allocated from a total of 700 million data points, which almost covered all landline telephone numbers in Taiwan. The representativeness of the final sample (age, sex, and geographic distribution) was examined by the chi-square goodness-of-fit, contrasted with the registered household statistics data that were provided and openly assessed by the Taiwan Ministry of Inferior. If there was a significant difference regarding the distribution of sex, age, and geographic characteristics between the final sample and the general population, weighting with the raking method (also known as iterative proportional fitting) was used to ensure that the demographic distribution of our sample accurately represented the general population. In total, 5430 respondents aged more than 15 years were contacted, and 2126 respondents agreed to take part in the anonymous telephone survey and completed the interview (sampling error of ±2.10% with a 95% confidence interval).

### 2.2. Measurements

The questionnaire was used to collect demographic information (age, gender, education level, occupation, and marital status), health-related measures (self-rated physical/mental health and self-efficacy), and suicide risk factors (suicide-related items, psychopathology, and substance use). The definitions and assessment strategies for these key variables are listed below.

#### 2.2.1. Prescription Drug Misuse (PDM)

A single question was used to assess whether the respondent had used prescription drugs that were not prescribed by physicians (e.g., sedatives, hypnotics, and analgesics) in the past year. The response was recorded as a self-reported binary response of yes or no. If the response was yes, further questions were asked to assess whether previous substance misuse led to dependency, withdrawal symptoms, or life impairments. This one-question format was used in a previous large-scale survey to elicit quick and reliable responses [21].

#### 2.2.2. Psychopathological Symptom Assessment

The five-item Brief Symptom Rating Scale (BSRS-5) was used to measure the level of psychological distress or psychopathological symptom severity of the respondents over the past week. The BSRS-5 is a 5-point Likert scale (with ratings from 0 to 4) and contains the following items [22]: 1. having trouble falling asleep (insomnia), 2. feeling tense or keyed up (anxious mood), 3. feeling easily annoyed or irritated (hostility), 4. feeling low in mood (depressed mood), and 5. feeling inferior to others (inferiority). The respondents were given brief and precise instructions before the description of symptoms to ensure validity and guide the participants in rating their degree of distress related to each item during the past week, including the current day. The total score ranged from 0 to 20, with higher scores indicating higher mental distress. The presence of psychiatric morbidity was defined as a BSRS-5 score of ≥ 6 [23]. The BSRS-5, either self-rated or administered through interviews, has been reported to have satisfactory psychometric properties (Cronbach’s alpha: 0.89) in detecting psychiatric morbidity and recent suicide ideation in medical settings and the community [23,24,25]. In this study, the Cronbach’s alpha was detected as 0.81.

#### 2.2.3. Suicide Risk Assessment and Help-Seeking Experience

Previous suicide ideation of the respondents was evaluated across different timespans (i.e., lifetime, past year, past month, and past week). In addition, the respondents were asked whether they had attempted suicide and, if so, when this occurred (i.e., lifetime, past year, and past month). Help-seeking experience was divided into four categories of services, including psychiatric services, non-psychiatric medical clinics, other mental health professionals, and folk therapy or religion.

#### 2.2.4. Health-Related Measures

A single question was used to assess health self-efficacy [26]. The participants were asked, “How much confidence, on a scale from 0 to 100, do you think you have regarding control over your own health conditions?”. Higher numbers indicate higher levels of confidence in perceived health control. The score of self-efficacy was also divided into three categories based on tertiles for analysis due to the non-normal distribution feature. The respondents were also asked to self-report physical and mental health conditions, with the ratings from 1 (very poor) to 5 (very good) [27].

### 2.3. Statistical Analysis

The data were analyzed after weighting for geographic distribution, age, and gender using a ranking method to ensure that the sample was representative of the general population. Missing values, outliers, and normality were checked prior to analysis to avoid violation of statistical principles. Variables that were not normally distributed were analyzed using a categorized format in descriptive or analytical statistics, such as help-seeking experience or health-related measures. In addition to descriptive statistics of demographic variables, the following tests were used. Univariate regression and bivariate regression were conducted as prerequisites for multivariate analyses. Associations between independent variables and PDM were assessed using chi-square tests and by calculating odds ratios (ORs) with 95% confidence intervals (CIs). To identify low health self-efficacy, the self-efficacy scale was split into tertiles, and the participants in the lower tertile were categorized as having low health self-efficacy. Multivariate logistic regression was applied to investigate the independent effects of demographics (e.g., age, gender, education), psychological distress, health self-efficacy, and lifetime suicide ideation or attempt on lifetime PDM in the model. All statistical analyses were performed using SPSS v21, and the significance level was set at *p* < 0.05.

## 3. Results

### 3.1. Personal Characteristics and Prevalence of PDM

We obtained available information on PDM during the past year from 2126 participants, of whom 1078 were women (50.7%). The majority of the participants were 25–44 years old (35.0%), and nearly equal numbers aged 45–64 years (34.2%). Most participants had an education level of college and above (54.3%) and were married (63.3%). The educational background was comparable to the national data (46.5%) of the Ministry of the Interior in the same year (2018), indicating the representativeness of this sample. The weighted 1-year prevalence of PDM was 8.5% (*n* = 180). Among those with PDM, 16.1% (*n* = 29) presented with dependency, 8.9% (*n* = 16) had withdrawal symptoms, and 3.3% (*n* = 6) had functional impairment. No significant difference was observed in the prevalence of PDM in the participants with different ages, marital statuses, or education levels; a borderline significant difference was observed between genders, with a female predominance (*p* = 0.06) (Table 1).

### 3.2. Associated Psychopathology and Psychosocial Factors of PDM

As shown in Table 2, individuals who reported PDM had significantly higher ratings for every item of the BSRS-5 (insomnia, anxiety, hostility, depression, and inferiority) and psychiatric morbidity (BSRS-5 score ≥ 6) than individuals who did not report PDM. Compared with the participants who did not report PDM, those who reported PDM also had a significantly higher prevalence of suicide ideation and suicide attempts at different timespans and were significantly more likely to seek psychiatric services (Table 2). Furthermore, individuals who misused prescription drugs had significantly lower self-rated mental and physical health as well as significantly lower self-efficacy scores than individuals who did not misuse prescription drugs (Table 3). To consider the confounding effects, all the abovementioned significant psychosocial and psychopathological correlates of PDM (e.g., demographic variables, self-efficacy, self-rated health, and individual items of the BSRS-5) were entered in the multivariate stepwise logistic regression analysis. The results revealed that insomnia (OR = 1.52), depression (OR = 1.77), lower (OR = 2.29) or medium level of self-efficacy (OR = 1.73), and lifetime suicide ideation (OR = 1.68) were significant correlates of PDM in the logistic model after controlling for age, gender, and education (Table 4).

## 4. Discussion

In this study, 8.5% of the participants reported misuse of prescription drugs (use of prescription drugs not prescribed by a physician) in the past year. The 1-year prevalence was similar to that reported by studies in the United States (1-year prevalence of Benzodiazepine and prescription opioid misuse: 2.2–5.8% and 4.1–4.7%, respectively) [28,29,30] and the European Union (1-year prevalence of Benzodiazepine and analgesics misuse: 5.8% and 5.0%, respectively) [31]. Because analgesics may also include non-opioid analgesics, the actual prevalence of PDM in Taiwan and the United States may be more similar. By contrast, the past-year prevalence of PDM we observed was significantly higher than that of the 2014 National Survey of Substance Use in Taiwan, which reported a weighted past-year prevalence of 3.7% for non-medical prescription drug use [32]. The difference between our study and the previous survey can be attributed to several explanations. One is that we did not recruit adolescents (aged 12–15 years) in our study. In the 2014 survey, the prevalence of PDM in adolescents was much lower than that in the general population. Additionally, we used different screening tools in this study. Given these possible explanations, the issue of PDM should be studied further because of its consequences and severity in recent years and in the future.

We did not include stimulants in our guiding sentence for the screening question of PDM. This is because prescription stimulant use was not well-established in the previous national survey in Taiwan because of its low incidence [16]. The reason for the low incidence of prescription stimulant misuse in Taiwan may be that the only stimulant approved as a prescription drug is methylphenidate. Amphetamine and dextroamphetamine, which are stimulants commonly used in other countries (e.g., the United States), have higher potential for abuse and are more likely to induce psychosis [33]; these are not approved for medical purposes by the Food and Drug Administration in Taiwan. However, the problematic use of stimulant prescriptions may be underestimated in Taiwan compared with other countries. In the United States, the past-year prevalence of stimulant misuse was 1.9% [34]. In a previous review that evaluated the non-medical use of prescription drugs in young adults, the prevalence rates of lifetime non-medical use of prescription stimulants varied from 6.9% to 19.8% [35]. These data indicate that prescription stimulant misuse may be a problem that warrants further investigation.

The majority of the respondents were aged 25–44 years (35.0%) in this study, which was compatible with the age group in the 2014 national survey in Taiwan. Similar results were observed in previous surveys in Japan [36] and the European Union [31], in which the major age group was in their 40s. By contrast, previous national surveys on drug use and health in the United States identified the younger population (aged 18–24 years) as the major population of PDM [8,16]. These differences indicate the importance of cultural consideration when interpreting PDM data in other countries. The prevalence of PDM did not differ significantly between the married population and the unmarried population. This finding conflicts with the finding of the previous national survey in Taiwan, which revealed that unmarried participants had a higher prevalence of buying nonprescription or over-the-counter drugs than those who were married [16]. However, people do not only obtain nonprescription drugs by purchasing them directly; they can also acquire them from friends, peers, family members, drug dealers, or purchases abroad [18]. Therefore, the previous survey may not present the actual difference in the prevalence of PDM between the married population and the unmarried population. In a systematic review of Benzodiazepine use [9], being unmarried was associated with Benzodiazepine misuse, but this finding was non-significant. We did not collect employment data in our study; however, these data have been analyzed in previous studies. In a Taiwan nationwide survey, employed individuals had a higher prevalence of buying sedatives and hypnotics, whereas unemployed individuals had a higher prevalence of buying analgesics [16]. Conversely, previous reports indicated unemployed individuals had a higher prevalence of using Benzodiazepines [8], whereas employed individuals had a higher prevalence of using prescription opioids [37]. This discrepancy suggests that PDM is similar to other types of substance misuse, with varying characteristics across different cultures or countries.

In this study, the BSRS-5 scores revealed that suicide ideation and suicide attempts were more common in the participants who reported PDM than in those who did not. Moreover, stepwise logistic regression indicated that depression and insomnia could significantly predict PDM. This implies that people with psychological distress may be more likely to misuse prescription drugs. This finding is in agreement with the findings of previous studies that people who misused prescription drugs presented more severe psychopathology; in a national epidemiological survey on alcohol and related conditions in the United States [9,11], any opioid or tranquilizer/sedative non-medical use of prescription drugs was associated with increased risks of alcohol and other substance use disorders as well as depressive, bipolar, and anxiety disorders. Dual pathology of substance use disorder and depression is very common in the field of addiction, and it can complicate treatment and worsen prognosis [38]. In addition, our findings support that higher suicidal ideation level was a significant correlate of any prescription drug misuse with higher odds (e.g., aOR = 1.22) [2,12]; further, they fill the knowledge gap in pointing out the role of PDM in the attribution to suicide risk in various timeframes spanning from the past week, month, year, to the lifetime. This finding warrants more attention for early suicide risk assessment among people with PDM.

Self-efficacy of individuals determines their coping strategies toward stress, their endurance when experiencing chronic or unsolvable physical or mental problems, and their capability of seeking help [26,39]. In this study, we used the concept of health-related self-efficacy to examine its association with PDM, and the result showed that it was an essential correlate of PDM. Previously there have been very few studies investigating this association; current literature was identified as related to PDM-avoidance self-efficacy or the influence of self-efficacy on alcohol misuse [39,40]. Moreover, one study implicated that poor academic self-efficacy is associated with non-medical prescription stimulant use among college students in the U.S. [41], while another study did not support the association between academic self-efficacy with misuse of prescription stimulants among graduate students [42]. Further, two studies revealed that self-efficacy is associated with the non-medical use of medication for mood enhancement [43,44]. Given the difference between terms used in studies, e.g., drinking refusal, self-efficacy, and health-related self-efficacy, our study findings provide another perspective to interpret prescription drug misuse providing a more influential role of perceived health control toward drug misuse behaviors. Under the cultural setting in Taiwan, the high healthcare accessibility and affordability of medication fees make it easy for the public to get a variety of prescribed medications through personal links to specialized services. In these circumstances, the condition and levels of perceived health coping or health-promoting behavior (i.e., health-related self-efficacy) might be influential to PDM. Our findings indicated that the lower the level of self-efficacy, the greater the tendency of drug misuse in the general public. More specifically, people with the lowest self-efficacy had approximately a two times higher probability of developing PDM. This informs the early identification of PDM high-risk groups and early intervention toward low self-efficacy to promote an appropriate medication consumption behavior.

To the best of our knowledge, this is the first large survey to demonstrate the association of PDM with self-efficacy and various psychopathological symptoms, including suicide risk. The study reflects the current conditions of PDM in the general population of Taiwan and emphasizes that medical practitioners should consider the possibility of misuse, addiction, or suicide risks when prescribing medications to patients with high levels of distress, insomnia, and depressive symptoms. In conclusion, PDM is a challenge in clinical practice that warrants more attention from medical providers. Individuals with PDM have a higher prevalence of insomnia, depression, and suicidality and tend to rely on help from non-psychiatric services because of their psychosomatic symptoms and lower health self-efficacy, which may increase national medical expenses. Psychoeducation interventions should be aimed toward individuals at high risk of PDM who seek help from non-psychiatric specialties, and these individuals should be screened for psychopathological symptoms and low self-efficacy and referred to appropriate mental health services for early management. Further research focusing on the influence of low self-efficacy on PDM, or vice versa, is needed to inform preventive strategies in the future.

The telephone interview design of this study has innate limitations. The research condition depended on the participants’ setting and may lead to selection or response bias. This was avoided through calling at proper times with structured interview questionnaires performed through trained interviewers and standardized procedures. Such interview methods may not reach people who do not use the landlines frequently, and the ages and genders of those who declined to respond were not recorded, thus limiting our comparison of respondents and non-respondents. Additionally, PDM was determined by a single question, so it may not represent a clinical diagnosis of substance use disorder or prescription drug abuse. The rates of PDM may also be underestimated because of false responses or the validity of the judgment of PDM, which may require further study design using clinical data to clarify. We did not differentiate the classes of misused drugs by self-report or examination of certain stimulants in the original research design; instead, we would like to collect a general experience of the public due to the high accessibility of prescription drugs and affordable health service expenses under national health insurance coverage in Taiwan. Further studies are suggested to focus on certain categories of prescription drugs (e.g., sedatives, analgesics, and stimulants) for specific implications. In addition, some other potential health cognitive/psychological factors that may play an important role in PDM, such as social norms, outcome expectancies, and perceived risk, could be considered for future studies. Given the limitations stated above, telephone respondents may speak more freely about psychopathological issues, including suicide. Although the cross-sectional design limits the inference of the causal effect, the findings contribute insights into the psychopathology and psychosocial features of the relatively novel field of PDM.

## 5. Conclusions

In this nationwide representative sample, individuals who misused prescription drugs had a higher prevalence of psychological distress and suicidality as well as lower levels of self-rated health. Early engagement of misused drug users who have comorbid psychological issues, including poor sleep or mental distress, is critically needed. Under the cultural context of high accessibility of healthcare services, primary care practitioners should consider assessing prescription drug misuse and focus on preventing further addictive behavior when prescribing medications for people with poor sleep, depressive symptoms, or low self-efficacy in maintaining their health conditions.

## Figures and Tables

**Table 1 ijerph-18-12961-t001:** Weighted prevalence of prescription drug misuse and related problems of different demographic groups (*n* = 2126).

	Total	Prescription Drug Misuse					Chi-Square *
		No	Yes	Dependency	Withdrawal	Functional Impairment	
	N	*n* (%)	*n* (%)	*n* (%)	*n* (%)	*n* (%)	*p*-Value
Gender							
Male	1048	972 (92.7)	77 (7.3)	9 (0.9)	3 (0.3)	4 (0.4)	0.06
Female	1078	974 (90.4)	103 (9.6)	20 (1.9)	13 (1.2)	2 (0.2)	
Age							
15–24	301	274 (91.1)	27 (8.9)	0 (0)	1 (0.5)	0 (0)	0.2
25−44	745	686 (92.2)	58 (7.8)	10 (1.3)	4 (0.6)	3 (0.4)	
45−64	728	655 (90.0)	73 (10.1)	13 (1.7)	8 (1.1)	3 (0.4)	
≥65	352	330 (93.6)	22 (6.4)	7 (1.9)	2 (0.5)	0 (0)	
Marriage							
Unmarried	704	647 (92.0)	57 (8.1)	6 (0.9)	4 (0.6)	3 (0.5)	0.1
Married	1346	1233 (91.6)	113 (8.4)	21 (1.6)	9 (0.7)	2 (0.2)	
Divorce	41	34 (83.9)	7 (16.1)	1 (2.1)	1 (2.2)	1 (2.2)	
Widowed	28	26 (93.4)	2 (6.6)	1 (3.8)	1 (3.8)	0 (0.0)	
Unknown	8	6 (73.1)	2 (26.9)	0 (0.0)	0 (0.0)	0 (0.0)	
Education							
Primary school	164	151 (92.4)	13 (7.6)	1 (0.9)	0 (0.3)	0 (0.0)	0.2
Junior high	222	202 (90.9)	20 (9.2)	7 (3.0)	2 (0.9)	0 (0.0)	
Senior	583	537 (92.2)	46 (7.8)	11 (1.8)	3 (0.5)	1 (0.2)	
College	294	267 (90.6)	28 (9.4)	1 (0.4)	2 (0.5)	0 (0.1)	
University	725	665 (91.7)	60 (8.3)	9 (1.3)	9 (1.2)	4 (0.5)	
Graduate	136	123 (90.5)	13 (9.5)	0 (0.0)	0 (0.0)	1 (0.6)	
Unknown	2	1 (50.0)	1 (50.0)	0 (0.0)	0 (0.0)	0 (0.0)	

* The Chi-square test was used to test the demographic difference between prescription drug misuse and non-PDM.

**Table 2 ijerph-18-12961-t002:** Psychopathology, suicidality, and help-seeking experience of individuals with different prescription drug misuse statuses (*n* = 2126).

*n* (%)	Prescription Drug Misuse		
	Yes	No	
	(*n* = 180)	(*n* = 1946)	*p*-Value
Psychopathology			
Insomnia	71 (39.4)	462 (23.7)	<0.0001
Anxiety	43 (23.9)	281 (14.4)	<0.001
Hostility	60 (33.3)	392 (20.1)	<0.0001
Depression	58 (32.2)	309 (15.9)	<0.0001
Inferiority	41 (22.8)	268 (13.8)	<0.0001
BSRS-5 ≥ 6	25 (13.9)	111 (5.7)	<0.0001
Suicide Ideation			
Lifetime	42 (23.3)	205 (10.5)	<0.0001
Past one year	9 (5.0)	33 (1.7)	<0.001
Past month	5 (2.8)	10 (0.5)	<0.05
Past week	6 (3.3)	22 (1.1)	<0.05
Suicide Attempt			
Lifetime	9 (5.0)	29 (1.5)	<0.001
Past year	1 (0.6)	0 (0.0)	<0.05
past month	1 (0.6)	0 (0.0)	<0.05
Help-Seeking Experience			
Psychiatric service	12 (6.7)	70 (3.6)	<0.05
Non-psychiatric medical service	11 (6.1)	77 (4.0)	0.2
Other mental health professionals	7 (3.9)	39 (2.0)	0.1
Folk therapy or religion	11 (6.1)	86 (4.4)	0.3

**Table 3 ijerph-18-12961-t003:** Self-efficacy and self-rated health of individuals with different prescription drug misuse statuses (*n* = 2126).

*n* (%)	Prescription Drug Misuse		
	Yes	No	
	(*n* = 180)	(*n* = 1946)	*p*-Value
Mental Health			
Very good	47 (26.1)	670 (34.5)	<0.001
Good	87 (48.3)	937 (48.2)	
Fair	28 (15.6)	273 (14.0)	
Poor	13 (7.2)	49 (2.5)	
Very poor	3 (1.7)	11 (0.6)	
Unknown	2 (1.1)	4 (0.2)	
Physical Health			
Very good	26 (14.4)	371 (19.1)	<0.001
Good	86 (47.8)	926 (47.6)	
Fair	36 (20.0)	467 (24.0)	
Poor	18 (10.0)	149 (7.7)	
Very poor	14 (7.8)	29 (1.5)	
Unknown	0 (0.0)	3 (0.2)	
Self-efficacy ^, Mean (SD)	74.2 (14.7)	79.9 (13.8)	<0.001
Low	83 (47.0)	585 (30.6)	<0.0001
Medium	64 (36.1)	691 (36.2)	
High	30 (16.9)	634 (33.2)	

^ The variable of health-related self-efficacy was measured by self-report on a scale of 0–100, in which 0 is the lowest point indicating the least confidence in perceived health control. It was categorized into tertiles in correlational statistics due to its non-normality feature.

**Table 4 ijerph-18-12961-t004:** Stepwise logistic regression on prescription drug misuse and associated variables (*n* = 2126).

	Prescription Drug Misuse	
	OR (95%CI)	*p*-Value
Gender		
Male	1	
Female	1.30 (0.95–1.79)	0.1
Age		
15–24	1.46 (0.80–2.66)	0.4
25–44	1.13 (0.67–1.89)	0.5
45–64	1.48 (0.90–2.44)	0.2
≥65	1	
Psychopathology (BSRS-5)		
Insomnia	1.52 (1.06–2.16)	<0.05
Depression	1.77 (1.22–2.57)	<0.05
Self-Efficacy		
Low	2.29 (1.46–3.60)	<0.001
Medium	1.73 (1.10–2.72)	<0.05
High	1	
Suicide ideation (lifetime) ^	1.68 (1.08–2.61)	<0.05
Suicide attempt (lifetime) ^	1.21 (0.52–2.84)	0.7

BSRS-5: The 5-item brief symptom rating scale. ^ The reference of suicide-related factors in the model was the non-presence of these behaviors.

## Data Availability

The dataset of this study will be made available upon request.

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
