# Peer review of "Prevalence and Correlates of Prescription Drug Misuse in a Nationwide Population Survey in Taiwan"

_ijerph, 2021, doi:10.3390/ijerph182412961_

Round 1

Reviewer 1 Report

Comments to authors

It’s my pleasure to have this opportunity to review the manuscript entitled “Prevalence and correlates of prescription drug misuse in a nationwide population survey in Taiwan”. This manuscript reported prevalence and psychosocial factors of PDM among 2126 Taiwanese people from a stratified randomized sample. This study has shown several strengths. The topic is significant given the detrimental consequences of PDM. Methodologies are rigorous. The manuscript is well-written. I believe that the manuscript would be further improved with authors’ consideration of my several concerns and comments shown in the following.

Introduction

Line 40: The definition of PDM may be improved by clarifying the classes of commonly used prescription drugs given prescription drugs could involve multiple categories and their risk/protective factors would vary (e.g., opioids vs. anxiolytics; stimulants vs. sedatives).

Line 40: Given a serious public health concern, it will be great if the authors can introduce epidemiological information (e.g., prevalence) of PDM from global literature as well as Taiwan research/Asian studies (e.g., China). To my knowledge, there have been lots of US studies and several Chinese studies that documented PDM in young adults and adolescents.

Line 55: Evidence that supports the study hypothesis of the association of suicide ideation and PDM seems to be missing.

Line 61: I appreciate that the authors highlighted the role of self-efficacy in PDM. However, clarifications are needed for the statements—what kind of self-efficacy did the authors talk about? Self-efficacy for health management or self-efficacy for medically use of prescription drugs? Or the sense of control for psychological distress management? Please note that, although these are all ‘self-efficacy’ but they play inconsistent roles in PDM.

Line 61: In addition to self-efficacy, substance use can be influenced by multiple psychosocial/cognitive factors (e.g., perceived susceptibility or perceived benefits) in line with health behavior theories (e.g., health belief model). Additional statements may be needed to explain why the authors particularly focused on self-efficacy. I suggest authors citing review studies/meta-analyses which suggested that self-efficacy was a robust factor for drug misuse relative to other cognitive factors. I see the authors have briefly addressed them in the discussion. But it will be great to mention this early so the readers could have a better sense of them.

Line 72: Given that one of study hypotheses was about PDM and physical health, introduction section would benefit from briefly reviewing literature on physical/somatic symptoms and their association with PDM.

Line 80: The association of stigma/literacy and medical help in benzodiazepine users would need evidence and citations.

Methods

Did the authors do attrition tests on demographics (age, gender, location, etc.) between respondents and these who did not respond? Such information may provide a better picture about representativeness of the sample.

Line 112: I wonder if respondents recognized sedatives or hypnotics? Did the interviewers provide their brand names? Given that the authors mentioned the impact of literacy in the introduction, this may be a concern for the assessment.

Did respondents report their common misused classes of drugs? It will be helpful to report prevalence across drug classes which can better depict the epidemic.

Line  122: Besides the Cronbach’s alpha from previous research, the authors should also provide this for the current study.

It seems that the analyses involved help seeking measures. However, their information is missing in the measurement section.

Statistical analysis: Prior to the analyses, did the authors screen the data in terms of missing, outliers, and normality (if applicable)? I ask this since these would affect the logistic regression models.

Statistical analysis: Why did authors categorize health self-efficacy instead of entering it as a continuous variable in the model? Please note that a continuous form of the factor can provide more information than a categorical measure. Was it because of the violation of normality? Or such a measure was supported by previous research? Please clarify.

Results

There are more than half of the sample that attained college—this seems to be subject to sampling bias.

Does Table 4 show the results after accounting for other psychological measures (i.e., anxiety, inferiority, and hostility)? These showed to be significantly associated with PDM in bivariate analyses, but their results in multivariate analyses were missing. It seems that they were removed from the model given a stepwise logistic regression. Please note that the stepwise method would have a higher threat to Type I error than hierarchical model. The authors would need to clarify and explain why they did so (e.g., multicollinearity?).

Line 179: I may avoid calling them as “predictors” given a cross-section study.

Discussion

Line 208: The authors explained that the study did not measure stimulant misuse. This is an important message which should be mentioned in the introduction. However, this may be a concern since prescription stimulants have been widely identified as the common misused medications in young adults, while Ritalin or Concerta (methylphenidates) are available in the Taiwan market.

Line 273: Again, some other potential health cognitive/psychological factors would also play an important role in PDM, such as social norms, outcome expectancies, and perceived risk. The discussion would benefit from briefly noting them as a direction for future research.

Line 295: Some additional limitations may need to be discussed: (1) selection bias, (2) response bias, (3) only one item per each psychological measure (e.g., depression).

Reviewer 2 Report

Many thanks for giving me the opportunity to review the revised manuscript entitled ”Prevalence and Correlates of Prescription Drug Misuse in a Nationwide Population Survey in Taiwan“.

This study investigated the prevalence of prescription drug misuse (PDM) and its associated psychopathology and psychosocial factors in the general population in Taiwan. The study findings showed that individuals who misused prescription drugs had a higher prevalence of psychological distress and suicidality and lower self-rated health or self-efficacy. The main disadvantage of this study is that the data were collected through a telephone survey. In addition, reading the submitted manuscript, several questions arose and drawbacks were noticed, which I recommend to fix.

Materials and Methods.

  1. Lines 85-95: It is desirable to describe in more detail the sampling procedure: how the sample size was estimated, how the proportionality of the sample according to gender, age, geographical location was assessed using telephone numbers. "(sampling error of 2.10% with a 95% confidence interval)" - for what prevalence rate of health outcome variable it was estimated?
  2. Lines 104-105: How was it found that respondents were able to distinguish "sedatives, hypnotics, and analgesics" from other medications?
  3. Lines 121-122: "The presence of psychiatric morbidity was defined as a BSRS-5 score of ≥6" - on the basis of which it was chosen?
  4. Lines 149-150: "The analysis plan has not been pre-registered on a publicly available platform." What did you mean by that sentence? Maybe you wanted to say the results of the study? Maybe it's an ethical issue, not a statistical one?

Results.

  1. Table 1: Include line "Total". Is the chi-square test addressed to PDM? Please indicate.
  2. Table 2 or Table 3: Include data for self-efficacy.
  3. Table 4 and lines 174-177: Were the data on suicidal attempts included in the multivariate logistic regression analysis? Why not? This is important because suicides are mentioned in the conclusions.

Discussion.

  1. The relationship between PDM and suicide needs to be discussed further.

References.

  1. Literature sources should be described according to the requirements of the journal (see Instructions for Authors).

Thank you for considering my opinion. I encourage authors to keep on working to improve the manuscript.

Author Response

Response to the reviewer 2’s comments

Manuscript ID: ijerph-1450709

Re: "Prevalence and Correlates of Prescription Drug Misuse in a Nationwide Population Survey in Taiwan"

Dear Editor/Reviewer 2

Thank you for giving us the chance to revise our work. In response to your opinions, we provided our point-to-point response as follows.

Reviewer 2

Many thanks for giving me the opportunity to review the revised manuscript entitled ”Prevalence and Correlates of Prescription Drug Misuse in a Nationwide Population Survey in Taiwan“.

This study investigated the prevalence of prescription drug misuse (PDM) and its associated psychopathology and psychosocial factors in the general population in Taiwan. The study findings showed that individuals who misused prescription drugs had a higher prevalence of psychological distress and suicidality and lower self-rated health or self-efficacy. The main disadvantage of this study is that the data were collected through a telephone survey. In addition, reading the submitted manuscript, several questions arose and drawbacks were noticed, which I recommend to fix.

[Author Response]

Thanks for the comments. We have revised the text according the questions and comments. 

Materials and Methods.

  1. Lines 85-95: It is desirable to describe in more detail the sampling procedure: how the sample size was estimated, how the proportionality of the sample according to gender, age, geographical location was assessed using telephone numbers. "(sampling error of 2.10% with a 95% confidence interval)" - for what prevalence rate of health outcome variable it was estimated?

[Author Response]

Thank you for your question. The sampling procedure is detailed below. Taiwan Suicide Prevention Center is a subordinate organization of the Ministry of Health and Welfare in Taiwan. This center has been operating for more than 16 years and regularly conducts an annual survey of suicide-related issues in the people of Taiwan, aged 15 years old and older. The Taiwan Suicide Prevention Center has outsourced this survey work to a professional poll center that has established 7 million data of landline telephone numbers. To ensure that the telephone numbers that are not registered could be sampled with equal probability, the computer-assisted telephone interview system adopts the random digit dialing (RDD) method. The RDD randomly replaces the last two digits of the existent telephone number databank to yield a total of 700 million data points, which almost cover all landline telephone numbers in Taiwan. The target sample size for each year is a-priori set as 1600. The number of participants that need to be interviewed in each geographic area is allocated by a stratified proportional randomization method, based on the distribution of population size, age, and sex in 22 different geographic areas in Taiwan. If the allocated number of interviewees in a specific geographic area is too few to be representative, at least 68 interviewees in this area are required. Eventually, the final collected sample reaches 2000 each year. The representativeness of the final sample (age, sex, and geographic distribution) is examined by the chi-square goodness-of-fit, contrasted with the registered household statistics data that are provided and openly assessed by the Ministry of Inferior in Taiwan. If there is a significant difference regarding the distribution of sex, age, and geographic characteristics between the final sample and the population in Taiwan, weighting with raking method (also known as iterative proportional fitting) is used to ensure that the demographic distribution of our sample accurately represents the population.

We have described the sampling procedure in more detail as addressed above in the section of Methods. Please refer to Page 2, Line 99-107.

  1. How was it found that respondents were able to distinguish "sedatives, hypnotics, and analgesics" from other medications?

[Author Response]

In this survey, we did not ask the participants to distinguish the types of prescription drugs. Instead, we’d like to survey the general misuse experience of the public in this national representative sample. We have highlighted in the Introduction section about the aim of this study and some limitations of our design in the final paragraph of the Discussion section.

[Added Contents] “We did not differentiate the classes of misused drugs by self-report or examine certain stimulants in the original research design; instead, we would like to collect a general experience of the public due to the high accessibility of prescription drugs and affordable health service expenses under national health insurance coverage in Taiwan. Further studies are suggested to focus on certain categories of prescription drugs (e.g., sedatives, analgesics, and stimulants) for specific implications.” (Line 309-313, page 6 at Discussion section)

  1. Lines 121-122: "The presence of psychiatric morbidity was defined as a BSRS-5 score of ≥6" - on the basis of which it was chosen?

[Author Response]

The cutoff point to determine the presence of psychiatric morbidity was based on our previous validity studies on the subjects of psychiatric outpatients and community residents. The following reference has been added in the text: Lee MB, Liao SC, Lee YJ, Wu CH, Tseng MC, Gau SF, et al. Development and verification of validity and reliability of a short screening instrument to identify psychiatric morbidity. J Formos Med Assoc 2003;102(10): 687-694. (Reference 23) Thanks for the reviewer’s advice.

  1. Lines 149-150: "The analysis plan has not been pre-registered on a publicly available platform." What did you mean by that sentence? Maybe you wanted to say the results of the study? Maybe it's an ethical issue, not a statistical one?

[Author Response]

Thanks for the reminder. We have deleted this irrelevant sentence.

Results.

  1. Table 1: Include line "Total". Is the chi-square test addressed to PDM? Please indicate.

[Author Response]

We have made it clear in the Note at the bottom of the table. (Line 461-462)

Q: Table 2 or Table 3: Include data for self-efficacy.

[Author Response]

Thanks for the advice. Since Table 2 presents the psychopathology, suicidality, and help-seeking experience of the participants, and Table 3 shows self-efficacy and self-rated health of the sample, they serve different purposes. However, we think that in Table 3 the association of PDM status and different levels of self-efficacy could be presented, so we have added that part in Table 3. Please check Table 3 for more details. We believe this would add values for scientific evidence and practical implications for future studies.

  1. Table 4 and lines 174-177: Were the data on suicidal attempts included in the multivariate logistic regression analysis? Why not? This is important because suicides are mentioned in the conclusions.

[Author Response]

Thank you for this opinion. We have added the statistics about suicide variables in Table 4.

Discussion.

  1. The relationship between PDM and suicide needs to be discussed further.

[Author Response]

Thanks for your suggestion. Apart from adding in reference related to PDM and suicide in the Introduction, we have enhanced our discussion about their relationship in the Discussion section.

[Added Contents]

In addition, our findings support that higher suicidal ideation level was a significant correlate of any prescription drug misuse with higher odds (e.g., aOR=1.22) [2,12]; further, they fill the knowledge gap in pointing out the role of PDM in the attribution to suicide risks in various time frame spanning from past-week, month, year, to the lifetime. This finding warrants for more attention for early suicide risk assessment among people with PDM.” (Line 259-264 in Page5)

References.

  1. Literature sources should be described according to the requirements of the journal (see Instructions for Authors).

[Author Response]

We have double checked and assured that the literature sources are correct.

Thank you for considering my opinion. I encourage authors to keep on working to improve the manuscript.

 [Author Response] Dear Reviewer 2, thanks for your advice and encouragement. We will keep on working on it.

Reviewer 3 Report

The article is interesting, I have no objections to the methodology. The authors are aware of the limitations of the chosen method, which they wrote about at the end of the paper. However, I have some remarks that may improve the quality of this paper.

In my opinion in the final part of the Discussion section much more recent studies than those cited by the authors should be used. I know that Albert Bandura's article is original, but there are much more recent new studies this theme. Furthermore, some of the references cited in lines 271 and 273 are studies from about 30 years ago. There have certainly been changes in three decades that should be noted. Please try to find more recent studies; there are quite a few.

In the Conclusions section, I would try to look for some other important conclusions. The authors wrote that "prescription drug abusers had a higher prevalence of psychiatric disorders and suicidal tendencies, as well as lower levels of self-rated health and self-efficacy." However, it is safe to assume that those with higher prevalence of disorders and suicidal tendencies, lower self-esteem and sense of efficacy are more likely to need pharmaceutical support. The authors wrote that "a cross-sectional design limits the inference of a causal effect." But in this case, it's just as likely that drug abuse can lead to a variety of disorders, as it is that the presence of disorders of any kind can be the cause of drug use and, in many cases, drug abuse. Pointing out to physicians, careful prescribing is a good conclusion, but perhaps consider some more preventative interventions, harm reduction, etc.?

Please check the numbering order of the references in the text and the index at the end of the manuscript. In my opinion, reference 17 (line 110) corresponds more closely to the article, which is numbered 18 in the References list at the end. Likewise, there is a footnote in text 18, which is numbered 19 in the References.

Please bring the References section in line with the guidelines from the Journal page (punctuation, abbreviations, bold, proper notations, etc.), As I indicated above - please check the correspondence of the reference numbers given in the text with the numbering in the References section.

Also, please remove the unnecessary legend of graphs and tables at the end.

Author Response

Response to the reviewer 3’s comments

Manuscript ID: ijerph-1450709

Re: "Prevalence and Correlates of Prescription Drug Misuse in a Nationwide Population Survey in Taiwan"

Dear Editor/Reviewer 3

Thank you for giving us the chance to revise our work. In response to your opinions, we provided our point-to-point response as follows.

Reviewer 3

The article is interesting, I have no objections to the methodology. The authors are aware of the limitations of the chosen method, which they wrote about at the end of the paper. However, I have some remarks that may improve the quality of this paper.

Q: In my opinion in the final part of the Discussion section much more recent studies than those cited by the authors should be used. I know that Albert Bandura's article is original, but there are much more recent new studies this theme. Furthermore, some of the references cited in lines 271 and 273 are studies from about 30 years ago. There have certainly been changes in three decades that should be noted. Please try to find more recent studies; there are quite a few.

[Author Response]

Thank you for pointing out this defect. We have carefully reviewed relevant articles and replaced the original contents with proper references. The reference number 40-53 has been deleted and replaced with reference 40-45. Please check the revised text in the final paragraph of page 5-6, Line 265-286 for details.

[Added contents]

Self-efficacy of individuals determines their coping strategies toward stress, the endurance when experiencing chronic or unsolvable physical or mental problems, and their capability of seeking help [27,40]. In this study we used the concept of health-related self-efficacy to examine its association with PDM, and the result showed that it was an essential correlate of PDM. Previously there has been very few studies investigating this association; current literature was identified related to PDM-avoidance self-efficacy or the influence of self-efficacy on alcohol misuse [40,41]. Moreover, one study implicated that poor academic self-efficacy is associated with non-medical prescription stimulant use among college student in the U.S.[42], while another study did not support the association between academic self-efficacy with misuse of prescription stimulant among graduate student [43]. Further, two studies revealed that self-efficacy is associated with non-medical use of medication for mood enhancement [44,45]. Given the difference between terms used in studies, e.g. drinking refusal self-efficacy and health-related self-efficacy, our study finding provided another perspective to interpret prescription drug misuse providing the more influential role of perceived health control toward drug misuse behaviors. Under the cultural setting in Taiwan, the high healthcare accessibility and affordability of medication fees make it easy for the public to get a variety of prescribed medications through personal link to specialized services. In this circumstances, the condition and levels of perceived health coping or health promoting behavior (i.e. health-related self-efficacy) might be influential to PDM. Our findings indicated that the lower the level of self-efficacy, the more tendency of drug misuse in the general public. More specifically, people with the lowest self-efficacy had about 2 times more probability of developing PDM. This informs for early identification of PDM high-risk groups and early intervention toward low self-efficacy to promote an appropriate medication consumption behavior.” (Line 265-286, page 5-6)

Q: In the Conclusions section, I would try to look for some other important conclusions. The authors wrote that "prescription drug abusers had a higher prevalence of psychiatric disorders and suicidal tendencies, as well as lower levels of self-rated health and self-efficacy." However, it is safe to assume that those with higher prevalence of disorders and suicidal tendencies, lower self-esteem and sense of efficacy are more likely to need pharmaceutical support. The authors wrote that "a cross-sectional design limits the inference of a causal effect." But in this case, it's just as likely that drug abuse can lead to a variety of disorders, as it is that the presence of disorders of any kind can be the cause of drug use and, in many cases, drug abuse. Pointing out to physicians, careful prescribing is a good conclusion, but perhaps consider some more preventative interventions, harm reduction, etc.?

[Author Response]

Thank you for providing this constructive opinion. It is noteworthy that the healthcare and national health insurance systems allow Taiwanese people to access to any specialized service for any perceived symptoms, so the potential of harmful use of prescription drugs (or even overdose) is a threat to civilians health status. For those with PDM, it was unclear whether lower self-esteem and sense of efficacy or higher suicide risk resulted in drug misuse behavior. However, the key information from this study finding was that PDM was essentially associated with the above-mentioned variables, which warrant for clinical attention. Thus we highlighted the cultural setting and its potential risks for attention. We have revised the Conclusion section to make it more reflective under the cultural context of Taiwan and more implicative for future reference. (Line 324-327)

Q: Please check the numbering order of the references in the text and the index at the end of the manuscript. In my opinion, reference 17 (line 110) corresponds more closely to the article, which is numbered 18 in the References list at the end. Likewise, there is a footnote in text 18, which is numbered 19 in the References.

[Author Response]

We have ensured that the reference numbers and the corresponding contents in the text are correct. Thank you for your reminder.

Q: Please bring the References section in line with the guidelines from the Journal page (punctuation, abbreviations, bold, proper notations, etc.). As I indicated above - please check the correspondence of the reference numbers given in the text with the numbering in the References section. Also, please remove the unnecessary legend of graphs and tables at the end.

[Author Response]

Thanks. We have corrected the reference format according to the guidelines. We also deleted the unnecessary expression for the Tables at the end of the file.

Round 2

Reviewer 1 Report

I think the authors did a good job with answering my questions. The revisions have greatly improved the manuscript. I have only a minor suggestion as shown below. After addressing this, I think the manuscript can be considered for the publication.

Line 99-106. The newly added contents may need to be rephrased using past tense to keep consistency in the paragraph.

Author Response

Dear Reviewer 1

We appreviate your suggestion of minor revision. As your advice, we have made required amendment. Please see the attched file for more details. We wish to thank you for your kind feedback and encouragement so we could improve our manuscript.

Best wishes,

Jenny Wu